# Reasoning Is Not a Race: When Stopping Early Beats Going Deeper

**Mohan Zhang**[1][*]  **Jiaxuan Gao**[1][*]  **Shusheng Xu**[2]   **Yi Wu**[1][†]
[1]IIIS, Tsinghua University    [2]Ant Group
zhangmohan70@gmail.com, jxwuyi@gmail.com

## Abstract

We study the use of Process Reward Models (PRMs) for guiding Long Chain-of-Thought (CoT) reasoning in large language models. Although PRMs deliver fine-grained feedback in standard tasks, PRM-guided beam search does not consistently outperform PRM-free approaches in long CoT reasoning. We trace this shortfall to a "step quality degradation"—the expected step quality shows concave behavior, yielding unimodal or monotonically declining trends. To counteract this, we propose **Z**-Score **G**uided **E**arly **S**topping (**ZGES**), which halts search at the detected quality peak using local PRM-reward z-scores. Across multiple math benchmarks and model scales, ZGES outperforms both standard PRM-guided beam search and the PRM-free methods. Ablation studies further highlight the advantages and robustness of ZGES's adaptive stopping mechanism.

## 1   Introduction

Process Reward Models (PRMs) play a crucial role in test-time search of reasoning tasks [1] for large language models (LLMs). Compared to the Outcome Reward Models (ORMs) [2] that focus solely on the correctness of the final answer, PRMs offer finer-grained supervision by providing stepwise rewards throughout the reasoning process. By offering feedback at each step, PRMs help guide the model's reasoning path toward the correct solution path. Recent works have trained PRMs through automatically annotated labels to guide LLMs in beam search or to re-rank multiple candidate outputs from the LLMs [3–6], resulting in improved performance compared to naive sampling.

Chain-of-Thought (CoT) [7] prompting has emerged as a powerful method to enhance the reasoning ability of LLMs by decomposing complex problems into intermediate steps. Building on this technique, the emergence of Long CoT LLMs capable of sustaining long-horizon reasoning chains, such as OpenAI o1 [8] and DeepSeek R1 [9], has sparked growing interest in long-CoT reasoning. These models demonstrate significant advances in mathematical reasoning, code synthesis, and cross-domain inference [10–12]. Compared to short CoT, Long CoT entails more comprehensive reasoning processes, greater depth in analyzing intermediate steps, and a wider exploration of logical relationships [12]. However, how to effectively apply PRMs to Long CoT reasoning tasks remains unclear. Specifically, we ask:

> **Q1:** *How well does PRM guide Long CoT models during test-time search?*

> **Q2:** *How to enhance PRMs' effectiveness in test-time search under Long CoT settings?*

In this work, to answer **Q1**, we evaluate the performance of Long CoT models with varying parameter sizes on math benchmarks including AMC2023, AIME2024, and AIME2025, using two prevalent classes of Process Reward Models: Hard PRM and Soft PRM (see Section 2). The hard PRM treats a

---

[*]Equal contribution.
[†]Corresponding author.

39th Conference on Neural Information Processing Systems (NeurIPS 2025).

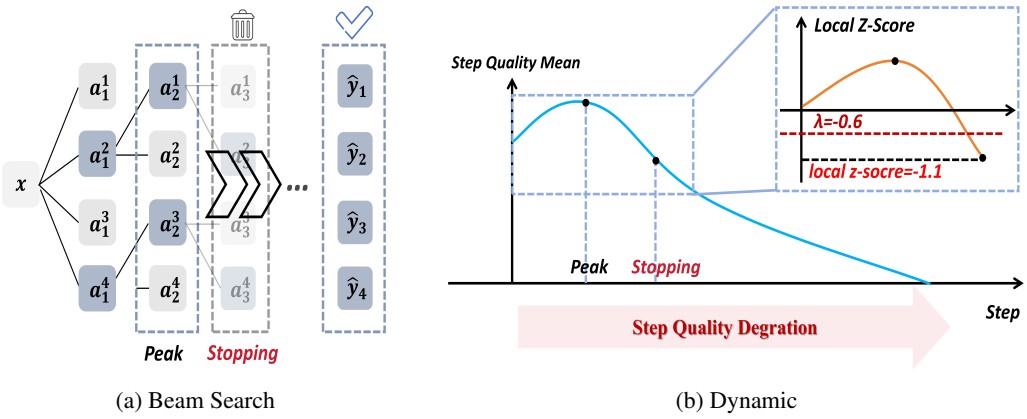

(a) Beam Search                                    (b) Dynamic

Figure 1: **Illustration of our proposed method: Z**-Score **G**uided **E**arly **S**topping for Beam Search (**ZGES**). The locally normalized z-score—which is calculated cumulatively based on the scores of all preceding and the current step—at the stopping point falls below $\lambda$ (due to step degradation), triggering early termination. The LLM policy then directly generates the remaining trajectory and final answer from the preceding step (i.e., the peak).

step as good if it can lead to the correct answer, while the soft PRM learns to predict the step's value under the LLM policy [3].

Our findings reveal that PRM-guided beam search does not consistently outperform PRM-free methods such as Majority@N in Long CoT reasoning. This outcome is counterintuitive, as beam search, guided by PRMs, is expected to explore and select intermediate reasoning steps more effectively than methods relying solely on ranking the full responses. These results suggest that the current implementation of PRM-guided beam search may not fully leverage its potential advantages in the context of Long CoT reasoning.

To investigate this phenomenon, we first perform in-depth analysis on the step quality $V$ during Beam Search—defined as the probability of reaching the correct final answer from any intermediate state—and observe that its trajectory is either unimodal or monotonically decreasing, a behavior we term "step quality degradation." We theoretically show that, due to PRM's degrading re-ranking ability over the course of Beam Search, the expected step quality typically exhibits concavity(see Proposition 3.8), providing a theoretical explanation for step quality degradation. These findings suggest that deeper reasoning steps in Beam Search are often suboptimal; we therefore propose early termination at the peak of expected step quality to preserve high-quality reasoning. Specifically, we introduce **Z**-Score **G**uided **E**arly **S**topping (**ZGES**, our answer to Q2), which terminates Beam Search based on the trend of step quality, identified by the local z-score of average PRM reward at the current step (see Figure 1). ZGES leverages the strong consistency between PRM rewards and true step quality in their z-normalized forms, thereby adaptively preventing overly deep reasoning and enabling efficient early stopping.

We evaluate ZGES on Long CoT models of varying scales, including DeepSeek-R1-Distill-Qwen-1.5B and 7B [9]. It consistently delivers strong performance across all Beam Search configurations and achieves the best results on all evaluated benchmarks.

Compared to standard beam search methods, ZGES achieves better performance with significantly lower computation requirement. For example, on the challenging AIME2024 benchmark, under the same expansion setting, ZGES with 16 beams outperforms standard Beam Search with 16 beams ($60.0\% \rightarrow 66.7\%$) when using R1-Distill-Qwen-1.5B. We also conduct an ablation study in Section 5, which includes a comparison between ZGES and fixed-step stopping to highlight the benefits of adaptive stopping, as well as a sensitivity analysis on the hyperparameter $\lambda$ to demonstrate the robustness of ZGES.

## 2   Preliminary

**Formalizing LLM reasoning as an MDP**    We model the reasoning process of LLMs as a Markov Decision Process (MDP) to enable formal analysis. A standard MDP is defined as a tuple $\mathcal{M} =$

Table 1: **Performance of different methods.** We consider settings with the same total budget $N = B \times E$ as equivalent, although Beam Search introduce additional computation overhead due to the use of a reward model. **Bold black numbers** indicate the best performance under equivalent settings. **Red superscripts** denote the performance difference of Beam Search compared to Majority@N under the same setting.

| Model | AMC23 | | | | | AIME24 | | | | | AIME25 | | | | |
|---|---|---|---|---|---|---|---|---|---|---|---|---|---|---|---|
| **(Majority@N)** | @8 | @16 | @32 | @64 | @128 | @8 | @16 | @32 | @64 | @128 | @8 | @16 | @32 | @64 | @128 |
| R1-Distill-Qwen-1.5B | 83.4 | 89.5 | 90.3 | 90.3 | 90.0 | 42.1 | 52.7 | 55.0 | 58.3 | 58.3 | 32.2 | 37.3 | 37.9 | 37.9 | 38.3 |
| R1-Distill-Qwen-7B | 94.0 | 95.0 | 96.0 | 96.0 | 97.0 | 68.0 | 77.0 | 78.0 | 78.0 | 76.7 | 50.0 | 56.0 | 56.0 | 56.0 | 56.0 |
| **($B \times E$)** | $4 \times 2$ | $8 \times 2$ | $16 \times 2$ | $32 \times 2$ | $64 \times 2$ | $4 \times 2$ | $8 \times 2$ | $16 \times 2$ | $32 \times 2$ | $64 \times 2$ | $4 \times 2$ | $8 \times 2$ | $16 \times 2$ | $32 \times 2$ | $64 \times 2$ |
| R1-Distill-Qwen-1.5B-Soft | 87.5 | 85.0$^{\downarrow 4.5\%}$ | 87.5$^{\downarrow 2.8\%}$ | 92.5 | **92.5** | 43.3 | 53.3 | 56.7 | 53.3$^{\downarrow 5.0\%}$ | 60.0 | 26.7$^{\downarrow 5.5\%}$ | 36.7$^{\downarrow 0.6\%}$ | 33.3$^{\downarrow 4.6\%}$ | 40.0 | 40.0 |
| R1-Distill-Qwen-1.5B-Hard | 92.5 | 87.5$^{\downarrow 2.0\%}$ | 92.5 | 92.5 | 90.0 | 36.7$^{\downarrow 5.4\%}$ | 40.0$^{\downarrow 12.7\%}$ | 53.3$^{\downarrow 1.7\%}$ | 56.7$^{\downarrow 1.6\%}$ | 63.3% | 33.3% | 30.0$^{\downarrow 7.3\%}$ | 36.7$^{\downarrow 1.2\%}$ | 36.7$^{\downarrow 1.2\%}$ | 36.7$^{\downarrow 4.1\%}$ |
| R1-Distill-Qwen-7B-Soft | 95.0 | 95.0 | 95.0$^{\downarrow 1.0\%}$ | 95.0$^{\downarrow 1.0\%}$ | **97.5** | 66.7$^{\downarrow 1.3\%}$ | 73.3$^{\downarrow 3.7\%}$ | 76.7$^{\downarrow 1.3\%}$ | 76.7$^{\downarrow 1.3\%}$ | **80.0** | 50.0 | 46.7$^{\downarrow 9.3\%}$ | 56.7 | 56.7 | **60.0** |
| R1-Distill-Qwen-7B-Hard | 95.0 | 92.5$^{\downarrow 2.5\%}$ | 97.5 | 95.0$^{\downarrow 1.0\%}$ | 95.0$^{\downarrow 2.0\%}$ | 63.3$^{\downarrow 4.7\%}$ | 70.0$^{\downarrow 7.0\%}$ | 73.3$^{\downarrow 4.7\%}$ | 76.7$^{\downarrow 1.3\%}$ | **80.0** | 53.3 | 43.3$^{\downarrow 12.7\%}$ | 56.7 | 60.0 | **60.0** |

$(\mathcal{S}, \mathcal{A}, T, r, \rho, H)$, where $\mathcal{S}$ is the state space, $\mathcal{A}$ is the action space, $T : \mathcal{S} \times \mathcal{A} \to \Delta(\mathcal{S})$ is the transition function, $r$ is the reward function, $\rho$ is the initial state distribution, and $H$ is the time horizon. A policy $\pi : \mathcal{S} \to \Delta(\mathcal{A})$ maps each state to a distribution over actions.

In the context of LLM reasoning, the input problem $x$ and the intermediate steps $\{a_1, a_2, \ldots, a_t\}$ define the state at step $t$ as $s_t = \{x, a_1, \ldots, a_t\}$. The action $a_{t+1} \sim \pi(\cdot \mid s_t)$ represents the next reasoning step. Since the next state is deterministically constructed by appending the chosen action to the current state, the transition function is deterministic: $s_{t+1} = T(s_t, a_{t+1}) = \{x, a_1, \ldots, a_t, a_{t+1}\}$. A trajectory $\tau = \{s_0, a_1, s_1, a_2, \ldots, s_T\}$ leads to a final answer, which is considered correct if it matches the ground truth $y$.

**Hard PRM and Soft PRM** While [1] obtains PRM annotations from human raters, this approach lacks scalability. More recent works [3, 5, 13] address this by training PRMs to predict automatically-generated annotations that estimate the future likelihood of solving the problem successfully. During training, PRMs can be categorized based on how they define step correctness, leading to two types: hard PRMs and soft PRMs [3]. Specifically, a hard PRM assumes that a reasoning step is good if it can eventually lead to the correct final answer, whereas a soft PRM evaluates the quality of a step based on the frequency with which it results in a correct solution. We denote a general process reward model as $r$, which maps a reasoning state $s$ to a real value in the range $[0, 1]$, i.e., $r : s \mapsto [0, 1]$. In particular, we define a hard PRM as $r_h : \mathcal{S} \to \{0, 1\}$ and a soft PRM as $r_s : \mathcal{S} \to [0, 1]$.

**PRM-guided Beam Search** Beam Search can unlock the potential of PRMs to assign credit to intermediate reasoning steps [4, 14, 15]. In this work, we denote the number of reasoning steps retained after pruning as the beam size (Abbreviated as Beam), and the number of expansions per Beam at each decoding step as the expansion factor(Abbreviated as Expansion). Specifically, we consider a Beam Search configuration with a fixed Beam size $B$ and expansion factor $E$, and follow the procedure below:

- From the initial state $s_0 = \{x\}$, sample $B \times E$ first-step candidates using the policy $\pi$, i.e., $\{a_1^{(i)} \sim \pi(\cdot \mid s_0)\}_{i=1}^{B \times E}$.
- score all $B \times E$ candidates using the PRM $r(\cdot)$, and select the top $B$ candidates based on their scores.
- From each of the selected $B$ candidates, generate $E$ next-step candidates using the policy $\pi$, resulting in $B \times E$ new candidates. Repeat this process iteratively.

Complete output trajectories from Beam Search are collected, and the final answer is chosen by Weighted Best of N (WBoN) [3], which selects the best result by weighting answer candidates based on the PRM rewards of the trajectories leading to these answers.

## 3 PRM Based Methods For Long COT Model

In this section, we answer **Q1**. We first compare PRM-guided Beam Search with Majority@N (Section 3.1), finding that PRM guidance does not improve performance. Then, in Section 3.2, we experimentally observe that during the Beam Search process, the step quality exhibits a unimodal

or monotonically decreasing pattern—referred to as step quality degradation—which underlies the suboptimal performance of Beam Search. Our theoretical analysis in Section 3.3 further confirms and explains this phenomenon, attributing it to the significant decline of PRM effectiveness as the search progresses.

## 3.1 Evaluate the Performance of PRM

**Evaluation Strategies and Baselines**  We primarily focus on evaluating the reasoning capability of PRM-guided Beam Search (introduced in Section 2). For comparison, we include PRM-free baseline Majority@N (i.e. self-consistency or majority voting).

**Experiment Setup**  Evaluations are performed on three challenging math benchmarks: **AMC2023**, **AIME2024**, and **AIME2025**. For PRM training, we follow the settings proposed by [3], including both the *hard* and *soft* configurations (details are provided in Appendix B.1). We instantiate our LLM policy with two Long CoT models of different parameter scales: **DeepSeek-R1-Distill-Qwen-1.5B** and **DeepSeek-R1-Distill-Qwen-7B** [9]. The base models used for PRM training are also aligned with these two variants, ensuring consistency between policy and reward modeling.

**Result**  As shown in Table 1, although the *Beam Search* achieves state-of-the-art accuracy under some equivalent settings, it performs worse than the *Majority@N*—which is PRM-free—in most cases when evaluated on the Long CoT models. This finding stands in contrast to previous work using non-Long CoT-based models [3, 4], where PRM-guided approaches generally achieved better performance than PRM-free baselines. This raises a key question: *what happens during the search process with Long CoT models?*

## 3.2 Degradation of Step Quality and Empirical Evidence

We define the step quality $V^\pi(s)$ as the probability that the LLM policy $\pi$ reaches the correct answer from the current state.

$$V^\pi(s) = p^\pi(\tau \mid s) \tag{1}$$

The step quality is equivalent to the state-value function in sparse-reward reinforcement learning, where $R(s, a) = 1$ upon task success and 0 otherwise [16]. The dynamics of $V(s)$ during Beam Search reflect the changing quality of explored reasoning paths. We thus empirically study step quality evolution in this process.

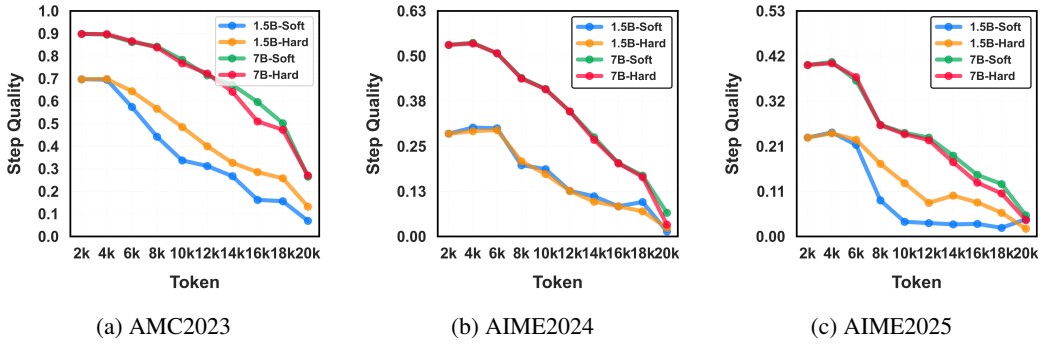

(a) AMC2023                    (b) AIME2024                    (c) AIME2025

Figure 2: **Empirical validation of Step Quality Degradation**

We used PRM to guide beam search under different configurations for the Long CoT model. We estimated the step quality of candidates using a **Monte Carlo** rollout method (details provided in Appendix B.2). The curves in Figure 2 are obtained by averaging the step quality of candidates at each search step over different configurations. The step quality consistently exhibits a monotonically decreasing or unimodal pattern as the number of steps increases (i.e., as $t$ grows).

We collectively refer to the two typical forms of the step quality—*unimodal* and *monotonically decreasing*—as *Step Quality Degradation*. Specifically, when the step index $t$ is sufficiently large (i.e., the reasoning involves many steps, which is **more common in long CoT models due to their extended output length**), the step quality tends to decrease to relatively low values. This degradation

indicates that Beam Search struggles to maintain high-quality candidates as $t$ increases during the reasoning process.

Furthermore, on two benchmarks (AMC2023 and AIME2025), beam search step quality under Hard PRM with the 1.5B model exceeds that under Soft PRM, while on all other settings they are comparable. Thus, Hard PRM appears more promising for guiding beam search. Consequently, all ZGES analyses and experiments in Section 4 use only the Hard PRM configuration.

## 3.3 Theoretical Analysis: Concavity of the Step Quality

In this section, we show in **Proposition 3.8** that the expected step quality is concave in $t$ due to PRM's degrading re-ranking ability in Beam Search, and provide a theoretical explanation for step quality degradation. The proof is derived under the extreme cases of two mild and empirically justifiable assumptions, Assumption 3.1 and Assumption 3.2.

**Assumption 3.1.** *For a reasoning-capable policy $\pi$, transitions from a correct state $s_t$ are more likely to yield a correct next state and final answer than those from an incorrect one, i.e., $p^\pi(s_{t+1} \mid s_t) \gg p^\pi(\bar{s}_{t+1} \mid s_t)$ and $p^\pi(\tau \mid s_t) > p^\pi(\tau \mid \bar{s}_t)$.*

In Assumption 3.1, we use the original letter to denote correctness, and a bar over the letter to indicate incorrectness. For instance, $s_t$ represents the correct state at step $t$, while $\bar{s}_t$ represents the incorrect state at step $t$. This assumption is similar to the one in [6], which has been empirically validated, so we will not elaborate further.

**Assumption 3.2.** *For a reasoning-capable policy and a well-aligned PRM, the PRM's re-ranking capacity gradually declines during Beam Search. Specifically, it becomes less effective in identifying the optimal candidate from the set $\left\{ a_t^{(i)} \sim \pi(\cdot \mid s_t) \right\}_{B \times E}$.*

Assumption 3.2 is based on our empirical observations, which will be validated in Appendix C.2. It indicates the failure of the PRM as Beam Search progresses. Let $P_{BS}^\pi(\bar{s}_{t+1} \mid s_t) = \epsilon_t$, which represents the probability of transitioning from the correct state $s_t$ to the incorrect state $\bar{s}_{t+1}$ during Beam Search (assuming Beam is 1 for simplicity; using other Beam sizes yields similar conclusions). Then we can obtain Corollary 3.3:

**Corollary 3.3.** *From Assumption 3.2, we can conclude that $\epsilon_t < \epsilon_{t+1}$, i.e., as the search progresses, it becomes increasingly likely to transition from a correct state to an incorrect one.*

A smaller $\epsilon_t$ implies that the PRM is more effective in selecting the optimal next step from the candidate set, so it reflects the re-ranking capability of the PRM. To capture the dynamic changes in step quality during the Beam Search process. Assuming the expectation of step quality:

$$E_t = P_{BS}^\pi(s_t|s_0)V^\pi(s_t) + P_{BS}^\pi(\bar{s}_t|s_0)V^\pi(\bar{s}_t) \tag{2}$$

It denotes the expectation of step quality at time step $t$ during the Beam Search process. We need to establish the variation pattern of $E_t$ throughout the Beam Search process. To facilitate the derivation, we consider an extreme case of Assumption 3.1, namely, for a reasoning-capable policy, we have $p^\pi(s_{t+1} \mid s_t) \to 1$ and $p^\pi(\bar{s}_{t+1} \mid \bar{s}_t) \to 1$. We can then state the following lemma (Proof in Appendix A.1):

**Lemma 3.4.** *Formally, when $p^\pi(s_{t+1} \mid s_t) \to 1$ and $p^\pi(\bar{s}_{t+1} \mid \bar{s}_t) \to 1$ for any t, we have $V^\pi(s_t) \to 1$ and $V^\pi(\bar{s}_t) \ll 1$.*

In this scenario, let $\Delta V_t$ denote the step quality difference between two consecutive correct states, where $\Delta V_t = V^\pi(s_{t+1}) - V^\pi(s_t)$. We have the following lemma showing that it is a small quantity:

**Lemma 3.5.** *Formally, when $p_\pi(s_{t+1} \mid s_t) \to 1$ and $p_\pi(\tau \mid s_t) \to 1$, it holds that $\Delta V_t \ll 1$.*

**Proof.** By applying Bayes decomposition, we can expand $V^\pi(s_t)$ as follows:

$$V^\pi(s_t) = p^\pi(s_{t+1} \mid s_t) \cdot V^\pi(s_{t+1}) + \underbrace{p^\pi(\bar{s}_{t+1} \mid s_t) \cdot V^\pi(\bar{s}_{t+1})}_{o(\delta^2)}$$

Therefore, $\Delta V_t = V^\pi(s_{t+1}) - V^\pi(s_t) \approx (1 - p^\pi(s_{t+1} \mid s_t)) \cdot V^\pi(s_{t+1}) = o(\delta) \ll 1$. $\square$

Similarly, we can prove that $\epsilon_t$ is also a small quantity (Appendix B.1). From this, we can obtain the second-order difference expression of the expectation of step quality $E_t$ in the following theorem:

**Theorem 3.6 (Second-order difference of $E_t$ ).** *Let $E_t = P_{BS}^{\pi}(s_t \mid s_0)V^{\pi}(s_t) + P_{BS}^{\pi}(\bar{s}_t \mid s_0)V^{\pi}(\bar{s}_t)$ denote the expectation of step quality. The second-order difference is given by $\Delta^2 E_t = E_{t+2} + E_t - 2E_{t+1}$. We conclude that the second-order difference of $E_t$ is:*

$$\Delta^2 E_t = p_{BS}^{\pi}(s_t \mid s_0) \left[ \underbrace{\Delta V_{t+1} - \Delta V_t}_{(A)} + \underbrace{V(s_t)(\epsilon_t - \epsilon_{t+1})}_{(B)} \right]$$

The proof of Theorem 3.6 is provided in Appendix A.3. We know from Corollary 3.3 that **term (B)** is strictly negative, while the sign of **term (A)** remains undetermined. Recall that $\epsilon_t$ is defined to characterize the re-ranking capability of the PRM at step $t$. So the absolute value of the term $V(s_t)(\epsilon_t - \epsilon_{t+1})$ (term B) serves as an indicator of how much the PRM's re-ranking accuracy deteriorates over time.

**Claim 3.7.** *Under Assumption 3.2, which states that PRM re-ranking accuracy **significantly** degrades during beam search (see the Appendix C.2), we expect $|\epsilon_t - \epsilon_{t+1}|$ to dominate $|\Delta V_{t+1} - \Delta V_t|$. Therefore, the second-order difference $\Delta^2 E_t = p_{BS}^{\pi}(s_t \mid s_0)\big(\Delta V_{t+1} - \Delta V_t + V(s_t)(\epsilon_t - \epsilon_{t+1})\big)$ is likely negative.*

Therefore, we state the following proposition:

**Proposition 3.8 (Concavity of expectation of step quality).** *Since the second-order difference $\Delta^2 E_t$ of the expectation of step quality $E_t = P_{BS}^{\pi}(s_t \mid s_0)V^{\pi}(s_t) + P_{BS}^{\pi}(\bar{s}_t \mid s_0)V^{\pi}(\bar{s}_t)$ is typically negative, $E_t$ is concave in the discrete sense.*

Proposition 3.8 shows that the expectation of step quality during the Beam Search process is generally concave, which results from the significant decline in PRM's re-ranking capability as the search progresses, as established in Claim 3.7.

We formalize a general property of concave sequences as follows:

**Lemma 3.9 (Degradation of Concave Sequences).** *If a discrete sequence $f_t$ satisfies $\Delta^2 f_t < 0$, then $\Delta f_t$ is non-increasing. In particular, when $f_t$ increases its growth slows down, and when $f_t$ decreases its decay accelerates.*

This concavity lemma directly accounts for the two observed patterns of step quality degradation:[3]

- *Unimodal* curves, where initial gains decelerate and eventually level off or reverse as growth slows;

- *Monotonic degradation*, where $\Delta E_t$ starts negative and keeps decreasing, leading to worsening quality over successive steps.

Table 2: Pearson correlation coefficients between PRM reward and step quality.

| Model | AMC2023 | AIME2024 | AIME2025 |
|---|---|---|---|
| R1-Distill-Qwen-1.5B-Hard | 0.991 | 0.977 | 0.935 |
| R1-Distill-Qwen-7B-Hard | 0.914 | 0.949 | 0.965 |

# 4 Z-Score Guided Early Stopping For Beam Search

We address **Q 2** as follows. Section 4.1 establishes the consistency of z-scores between PRM Reward and Step Quality, forming the basis of our method. Section 4.2 introduces our heuristic method ZGES, and Section 4.3 reports experimental results demonstrating its effectiveness.

---

[3]These two scenarios are theoretical conjectures characterizing common suboptimal behaviors of $E_t$. Concavity alone does not guarantee the function is strictly unimodal or monotonically decreasing.

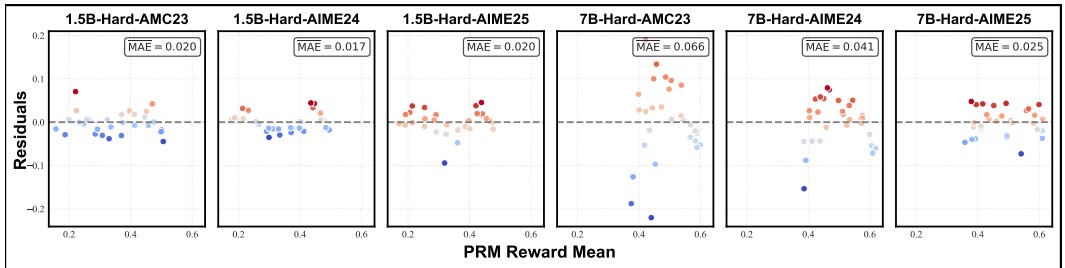

Figure 3: **Residual Analysis:** Linear fitting of mean PRM reward and step quality for candidates at each step under the same beam search settings, with MAE mean calculation.

## 4.1 Consistency of Z-Scores between PRM Reward and Step Quality

As established in Section 3, step quality degradation is a primary factor limiting the effectiveness of beam search in Long CoT reasoning tasks. This observation suggests a simple yet effective strategy: ***terminate the beam search at the point where the average step quality of candidates peaks, and let the LLM policy complete the remainder of the trajectory.*** Theoretically, this approach can maximize the expected likelihood that the final output includes a correct solution.

However, a key challenge is that PRM estimates—the only available signal during search—are evidently biased [17]. This calls for reinterpreting PRM scores and identifying a variable that better approximates true step quality. Notably, the average PRM reward of candidates exhibits a strong linear correlation with their average step quality (see Table 2 for Pearson correlation coefficients across beam search configurations on the same benchmark).

As observed, the Pearson correlation coefficients between them are consistently greater than $0.91$, indicating a strong linear correlation. To further enhance our judgment, we performed residual analysis on the mean PRM reward and mean step quality under different beam search settings. Additionally, we calculated the mean MAE for each beam search configuration. The results are presented in Figure 3. The residual plots and low MAE confirm a good linear fit between the two variables.[4]

Having established the linear relationship, we further show that the z-scores of the two variables remain consistent throughout the search process.

**Lemma 4.1** (Z-score Consistency). *Let $x_t$ and $y_t$ denote the mean PRM reward and mean step-quality of all beam-search candidates at step $t$ under a fixed beam setting $B \times E$:*

$$x_t = \tfrac{1}{B \times E} \sum_{i=1}^{B \times E} r\big(s_{t-1}, a_t^{(i)}\big), \quad y_t = \tfrac{1}{B \times E} \sum_{i=1}^{B \times E} V\big(s_{t-1}, a_t^{(i)}\big),$$

*where $\{a_t^{(i)} \sim \pi(\cdot \mid s_{t-1})\}_{i=1}^{B \times E}$ are the beam candidates. If there exist constants $k > 0$ and $b$ such that $x_t = k\, y_t + b$ for all t, then their z-normalized forms*

$$x_t' = \frac{x_t - \mu_x}{\sigma_x}, \quad y_t' = \frac{y_t - \mu_y}{\sigma_y}$$

*satisfy $x_t' = y_t'$ for all t. Hence, the z-scores of the mean PRM reward and the mean step-quality coincide throughout the search trajectory.*

***Proof.*** Since $x = ky + b$, we have

$$x' = \frac{x - \mu_x}{\sigma_x} = \frac{k\,y + b - \mu_x}{\sigma_x} = \frac{k\,(y - \mu_y)}{k\,\sigma_y} = \frac{y - \mu_y}{\sigma_y} = y',$$

which proves the claim. □

## 4.2 Method: ZGES

Section 4.1 shows that, for the Long CoT model, the average PRM reward and the average step quality of candidates at each beam search step exhibit strong alignment after applying z-normalization. Since

---

[4]Residuals are randomly scattered around zero without any clear pattern, indicating a good linear fit.

z-scores reflect relative deviations from the local mean, they provide a principled way to monitor the dynamics of search: a negative z-score at the final step, for instance, indicates a local decline in quality relative to previous steps. This motivates the use of locally computed z-scores as a signal for identifying potential performance drops during search. Our method can be formally described as follows.

1. Set the threshold $\lambda$ and Beam Search configuration $B \times E$.
2. Run Beam Search. At each step $t$, record the mean PRM reward of all candidates, $x_t$.
3. Compute the local z-score of $x_t$ using the sequence $\{x_i\}_{i=1}^t$.
4. If $x_t' < \lambda$, stop Beam Search early and directly decode from the candidates at step $t-1$; otherwise, continue from Step 2.

Figure 1 illustrates the proposed ZGES method. The hyperparameter $\lambda$ governs the sensitivity of our dynamic early-stopping criterion—intuitively, a smaller $\lambda$ leads to later termination steps, while a larger $\lambda$ results in earlier stopping. In Section 5.1, we conduct a sweep over different $\lambda$ values and observe that ZGES performance remains stable across a wide range of settings. This insensitivity to $\lambda$ highlights the robustness of our method.

### 4.3 Experiment

**Experiment Setup**  The experimental setup is consistent with Section 3.1. We compare our method with Majority@N and the standard PRM-guided beam search methods, including Hard PRM and Soft PRM. Our method requires only simple modifications to the beam search algorithm. As noted in Section 3.3, we only report results for our method using the Hard PRM setting in this section. In all Beam Search settings, the expansion factor is fixed to 2. For the hyperparameter $\lambda$, we set $\lambda = -0.6$ for the 1.5B model and $\lambda = 0$ for the 7B model. As shown in Section 5.1, the performance is generally insensitive to the choice of $\lambda$.

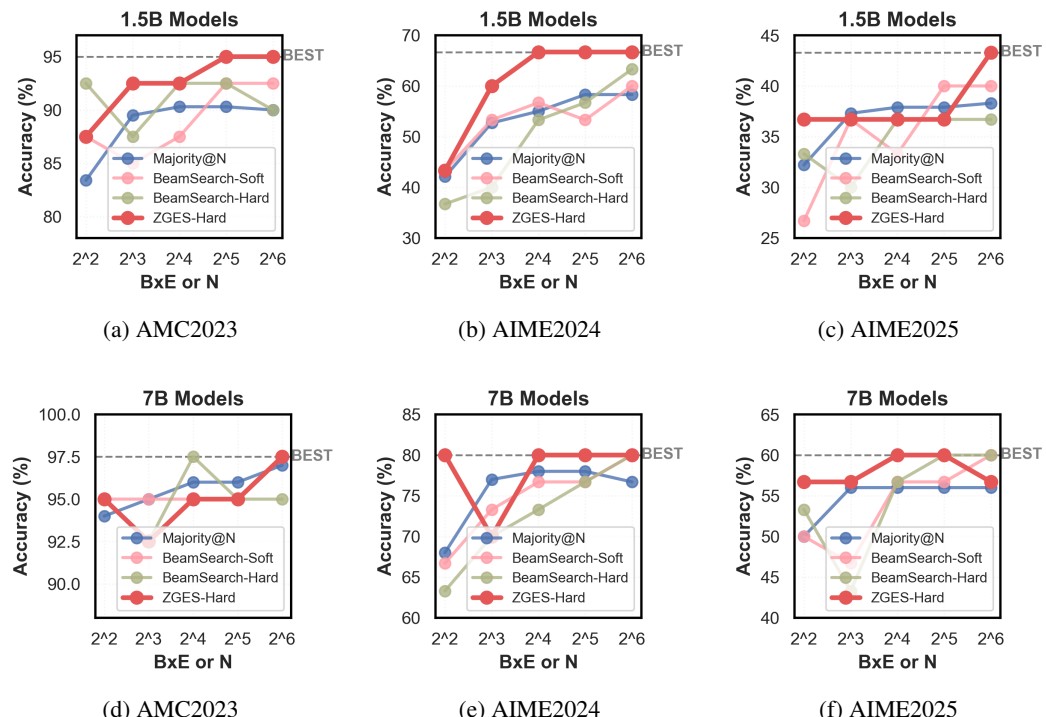

Figure 4: **Model Performance Results.** The term *B×E or N* denotes the product of beam size and expansion in Beam Search, or the value of $N$ in Majority@N, respectively.

**Main Result**  The comparison results are presented in Figure 4. **ZGES** demonstrates highly competitive performance compared to the baselines, with the following notable advantages:

- **Best Performance Across Settings.** Under all model scales and benchmark configurations, ZGES consistently achieves best results.

- **Robustness to Beam Size Scaling.** As the beam size increases (from 4 to 64), ZGES outperforms all other baselines in most scenarios, indicating strong robustness.

- **Improved Performance with Reduced Computational Cost.** Compared to standard beam search methods, ZGES achieves best performance even with relatively small beam sizes (as indicated by the intersection of the ZGES curve and the best performance line in the figures). Moreover, by incorporating an early stopping mechanism, ZGES reduces the number of PRM calls required during beam search (Table 3), effectively lowering computational overhead while maintaining—or even enhancing—generation quality.

## 5 Ablation Study

In this section, we conduct further analysis and discussion on the proposed ZGES method. In Section 5.1, we conducted hyperparameter sensitivity experiments of $\lambda$ to investigate its impact. Section 5.2 compares the behavioral and performance differences between ZGES and the fixed-step stopping beam search.

### 5.1 The performance of ZGES exhibits robustness to changes in hyperparameter.

The hyperparameter $\lambda$ is the sole tunable parameter in our method, ZGES. We analyze how variations in $\lambda$ affect the performance of ZGES. Specifically, we consider four values: $-0.4$, $-0.6$, $-0.8$, and $-1.0$. For each $\lambda$, ZGES is evaluated using multiple $B \times E$ Beam Search configurations, consistent with those in Sections 3.1 and 4.3.

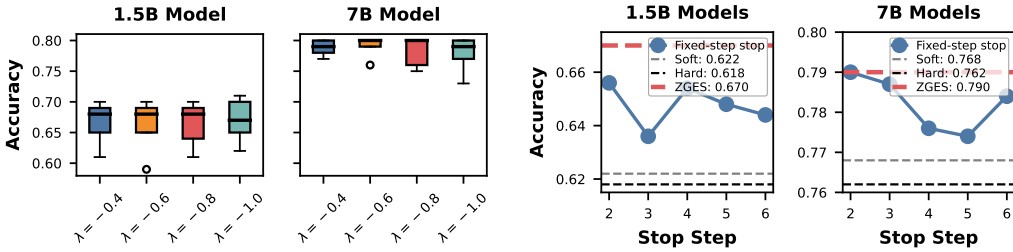

Figure 5: $\lambda$ Sensitivity Analysis      Figure 6: Comparison with Fixed-Step Stopping

The overall performance is assessed across three benchmark tasks. As shown in Figure 5, the results are visualized using box plots to illustrate the impact of different $\lambda$ values on the performance of ZGES. We observe that the performance of ZGES remains consistent across different choices of $\lambda$, indicating that the method is robust to the selection of this hyperparameter.

### 5.2 Dynamic Stopping is Better than Fixed-Step Stopping

To further highlight the advantage of ZGES with dynamic stopping, we compare it against a fixed-step stopping baseline that terminates Beam Search at a predetermined step.

As shown in Figure 6, both methods are evaluated under the Hard PRM setting. The reported results correspond to the average performance across three benchmark tasks, using the same Beam Search configurations as described in Section 5.1. In the figure, "Soft" and "Hard" denote Beam Search guided by the standard Soft PRM and Hard PRM settings, respectively. The results show that ZGES outperforms the fixed-step stopping baseline configured within a selected step range that is expected to contain high-quality steps (as suggested by Figure 2), which we used to reduce computational cost. Moreover, both methods consistently outperform standard Beam Search, which is expected since the degradation of step quality over time has been demonstrated in Section 3.

Table 3: **Comparison of PRM calls and token ratios across models and tasks.** 1.5B and 7B are the R1-Distill-Qwen models; ZGES(x) denotes ZGES with $\lambda = x$. PRM Call shows the average number of PRM calls; Tokens Ratio indicates the average number of tokens relative to Beam Search.

| Model | Method | AMC23 PRM Call | AMC23 Tokens Ratio | AIME24 PRM Call | AIME24 Tokens Ratio | AIME25 PRM Call | AIME25 Tokens Ratio |
|---|---|---|---|---|---|---|---|
| 1.5B | ZGES (-0.4) | **3.08 ↓61%** | **0.753** | **3.27 ↓66%** | **0.929** | **3.33 ↓65%** | **0.926** |
| 1.5B | ZGES (-0.8) | **3.18 ↓60%** | **0.763** | **3.42 ↓64%** | **0.956** | **3.48 ↓63%** | **0.942** |
| 1.5B | Beam Search | 7.91 | — | 9.55 | — | 9.53 | — |
| 7B | ZGES (-0.4) | **2.97 ↓56%** | **0.675** | **3.31 ↓63%** | **0.839** | **3.33 ↓62%** | **0.905** |
| 7B | ZGES (-0.8) | **3.17 ↓53%** | **0.692** | **3.54 ↓61%** | **0.856** | **3.58 ↓59%** | **0.911** |
| 7B | Beam Search | 6.69 | — | 9.00 | — | 8.73 | — |

## 5.3 Token Usage and PRM Calls

We further analyze the efficiency of ZGES in terms of PRM calls and token usage across different models and tasks (Intuitively, these two metrics will decrease compared to the standard method). Table 3 summarizes the results for the 1.5B and 7B models, comparing ZGES with standard PRM-guided Beam Search under different $\lambda$ settings.

From the table, we make three key observations. First, ZGES significantly reduces the number of PRM calls compared to the standard Beam Search (by more than 50% in most cases) while maintaining or even improving performance. Second, for both model sizes, smaller $\lambda$ result in more PRM calls, consistent with the analysis in C.1: smaller $\lambda$ delays the stopping point, leading to increased PRM usage. Third, the token ratio demonstrates that ZGES also reduces the overall token generation compared to standard Beam Search, indicating improved efficiency in computation and resource usage. Overall, these ablation results further validate the practical advantages of ZGES in both effectiveness and efficiency.

## 6 Conclusion

In this paper, we uncover a "step quality degradation" in long-horizon Beam Search—its step quality follows *unimodal* or *monotonic degradation* trends. To address this, we propose Z-Score Guided Early Stopping (ZGES), which monitors the local z-score of PRM rewards to detect the quality peak and terminate search before degradation sets in. Empirical evaluation on diverse math benchmarks and Long CoT models shows that ZGES consistently surpasses both standard Beam Search and PRM-free approaches, cutting computational cost while delivering stronger performance. Our work thus bridges theoretical insights into PRM limitations with a practical, efficient search strategy.

**Limitations** Due to limited computational resources, we could not explore a broader range of $\lambda$ values (e.g., more aggressive early stopping with $\lambda \in [0, 1]$). However, this does not hinder the demonstration of ZGES's effectiveness. In future work, we plan to investigate more flexible early stopping strategies beyond local Z-normalization.

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

# A  Theory Proof

## A.1  Proof of Lemma 3.4

**Lemma 3.4.** *Formally, when $p^\pi(s_{t+1} \mid s_t) \to 1$ and $p^\pi(\bar{s}_{t+1} \mid \bar{s}_t) \to 1$ for any t, we have $V^\pi(s_t) \to 1$ and $V^\pi(\bar{s}_t) \ll 1$.*

**proof.** Assume the reasoning horizon is $T$.

Then we have

$$p^\pi(\tau \mid s_t) \geq \prod_{i=t}^{T-1} p^\pi(s_{i+1} \mid s_i) \to 1.$$

therefore

$$V^\pi(s_t) = p^\pi(\tau \mid s_t) \to 1.$$

Similarly, for the error states, we have

$$p^\pi(\bar{\tau} \mid \bar{s}_t) \geq \prod_{i=t}^{T-1} p^\pi(\bar{s}_{i+1} \mid \bar{s}_i) \to 1,$$

which implies

$$V^\pi(\bar{s}_t) = 1 - p^\pi(\bar{\tau} \mid \bar{s}_t) \to 0.$$

$\square$

## A.2  Probability consistency between $P_{\text{BS}}^\pi$ and $p^\pi$

Under an extreme case of Assumption 3.1, where $p^\pi(s_{t+1} \mid s_t) \to 1$ and $p^\pi(\bar{s}_{t+1} \mid \bar{s}_t) \to 1$. We show that the Beam Search transition probabilities satisfy

$$P_{\text{BS}}^\pi(s_{t+1} \mid s_t) \to 1 \quad \text{and} \quad P_{\text{BS}}^\pi(\bar{s}_{t+1} \mid \bar{s}_t) \to 1.$$

Consider the special case in Section 3.3 where the beam is set to 1 to simplify the probability formulation of Beam Search. When starting from a correct state $s_t$, the candidate set $\{a_1^{(i)} \sim \pi(\cdot \mid s_t)\}_{i=1}^{B \times E}$ is sampled from the output distribution of $\pi$. If the next step is randomly sampled from the candidate set $\{a_1^{(i)} \sim \pi(\cdot \mid s_t)\}_{i=1}^{B \times E}$, then according to resampling theory, the selected action still follows the probability distribution induced by $\pi$. Furthermore, since PRM is generally more efficient than naive random sampling, we have

$$P_{\text{BS}}^\pi(s_{t+1} \mid s_t) \to 1.$$

Following the same reasoning, we also obtain

$$P_{\text{BS}}^\pi(\bar{s}_{t+1} \mid \bar{s}_t) \to 1.$$

Therefore, we have $\epsilon_t = P_{\text{BS}}^\pi(\bar{s}_{t+1} \mid s_t) \ll 1$.

## A.3 Proof of Theorem 3.6

**Theorem 3.6** (**Second-order difference of** $E_t$ ). *Let* $E_t = P_{BS}^\pi(s_t \mid s_0)V^\pi(s_t) + P_{BS}^\pi(\bar{s}_t \mid s_0)V^\pi(\bar{s}_t)$ *denote the expectation of step quality. The second-order difference is given by* $\Delta^2 E_t = E_{t+2} + E_t - 2E_{t+1}$. *We conclude that the second-order difference of* $E_t$ *is:*

$$\Delta^2 E_t = p_{BS}^\pi(s_t \mid s_0) \left[ \underbrace{\Delta V_{t+1} - \Delta V_t}_{(A)} + \underbrace{V(s_t)(\epsilon_t - \epsilon_{t+1})}_{(B)} \right]$$

**proof.** As mentioned in Section 3.3, we consider a special case based on Assumption 3.1, where the policy has strong reasoning ability. In this case, $p_\pi(s_{t+1} \mid s_t) \to 1$ and $p^\pi(\bar{s}_{t+1} \mid \bar{s}_t) \to 1$. We first identify several small terms: $\Delta V_t \ll 1$, $\epsilon_t \ll 1$, $V^\pi(\bar{s}_t) \ll 1$, $P_{BS}^\pi(\bar{s}_{t+1} \mid s_t) \ll 1$ and $P_{BS}^\pi(s_{t+1} \mid \bar{s}_t) \ll 1$ .(Appendix A.2, Lemma 3.4, Lemma 3.5)

In the derivation, we ignore the products of small terms, e.g., $\Delta V_t \cdot \epsilon_t$.

According to the definition of the expectation of step quality, we have:
$$E_t = P_{BS}^\pi(s_t \mid s_0)V^\pi(s_t) + P_{BS}^\pi(\bar{s}_t \mid s_0)V^\pi(\bar{s}_t)$$
$$E_{t+1} = P_{BS}^\pi(s_{t+1} \mid s_0)V^\pi(s_{t+1}) + P_{BS}^\pi(\bar{s}_{t+1} \mid s_0)V^\pi(\bar{s}_{t+1})$$
Therefore, the first-order difference of $E_t$, denoted as $\Delta E_t$, is given by:

$\Delta E_t = E_{t+1} - E_t$

$= [P_{BS}^\pi(s_{t+1} \mid s_0)V^\pi(s_{t+1}) - P_{BS}^\pi(s_t \mid s_0)V^\pi(s_t)] +$

$$\left[ \underbrace{P_{BS}^\pi(\bar{s}_{t+1} \mid s_0)V^\pi(\bar{s}_{t+1}) - P_{BS}^\pi(\bar{s}_t \mid s_0)V^\pi(\bar{s}_t)}_{o(\delta^2)} \right]$$

$$= \left[ P_{BS}^\pi(s_{t+1} \mid s_t)P_{BS}^\pi(s_t \mid s_0) + \underbrace{P_{BS}^\pi(s_{t+1} \mid \bar{s}_t)P_{BS}^\pi(\bar{s}_t \mid s_0)}_{o(\delta^2)} \right] V^\pi(s_{t+1}) - P_{BS}^\pi(s_t \mid s_0)V^\pi(s_t)$$

$= P_{BS}^\pi(s_t \mid s_0) [P_{BS}^\pi(s_{t+1} \mid s_t)V^\pi(s_{t+1}) - V^\pi(s_t)]$

$= P_{BS}^\pi(s_t \mid s_0) [(1 - \epsilon_t)(V^\pi(s_t) + \Delta V_t) - V^\pi(s_t)]$

$$= P_{BS}^\pi(s_t \mid s_0) \left[ \Delta V_t - \epsilon_t V^\pi(s_t) - \underbrace{\epsilon_t \Delta V_t}_{o(\delta^2)} \right]$$

$= P_{BS}^\pi(s_t \mid s_0) [\Delta V_t - \epsilon_t V^\pi(s_t)]$

Based on this, we derive the second-order difference $\Delta^2 E_t$ as follows:

$\Delta^2 E_t$

$= (E_{t+2} - E_{t+1}) - (E_{t+1} - E_t)$

$= \Delta E_{t+1} - \Delta E_t$

$= P_{BS}^\pi(s_{t+1} \mid s_0) [\Delta V_{t+1} - \epsilon_{t+1}V^\pi(s_{t+1})] - P_{BS}^\pi(s_t \mid s_0) [\Delta V_t - \epsilon_t V^\pi(s_t)]$

$= [P_{BS}^\pi(s_t \mid s_0)P_{BS}^\pi(s_{t+1} \mid s_t) + P_{BS}^\pi(\bar{s}_t \mid s_0)P_{BS}^\pi(s_{t+1} \mid \bar{s}_t)] [\Delta V_{t+1} - \epsilon_{t+1}V^\pi(s_{t+1})]$
$$- P_{BS}^\pi(s_t \mid s_0) [\Delta V_t - \epsilon_t V^\pi(s_t)]$$

$= P_{BS}^\pi(s_t \mid s_0) \{P_{BS}^\pi(s_{t+1} \mid s_t) [\Delta V_{t+1} - \epsilon_{t+1}V^\pi(s_{t+1})] - [\Delta V_{t+1} - \epsilon_{t+1}V^\pi(s_{t+1})]\}$
$$- P_{BS}^\pi(\bar{s}_t \mid s_0) \underbrace{P_{BS}^\pi(s_{t+1} \mid \bar{s}_t) [\Delta V_{t+1} - \epsilon_{t+1}V^\pi(s_{t+1})]}_{o(\delta^2)}$$

$= P_{BS}^\pi(s_t \mid s_0) \{(1 - \epsilon_t) [\Delta V_{t+1} - \epsilon_{t+1}(V^\pi(s_t) + \Delta V_t)] - [\Delta V_{t+1} - \epsilon_{t+1}(V^\pi(s_t) + \Delta V_t)]\}$

$$= p_{BS}^\pi(s_t \mid s_0) \left[ \underbrace{\Delta V_{t+1} - \Delta V_t}_{(A)} + \underbrace{V(s_t)(\epsilon_t - \epsilon_{t+1})}_{(B)} \right]$$

□

## B  Details of Experimental Setup

### B.1  PRM Training

To train hard PRM and soft PRM, we gather 9K challenging questions from the training data of the DeepScaleR project [18] by keeping problems that have non-trivial accuracy for DeepSeek-R1-Distill-Qwen-7B. For each problem, we generate 16 responses, and generate 8 completions for each step in the generated responses to construct the PRM training dataset. We follow [3] to construct the datasets for Hard PRM and Soft PRM training.

### B.2  Estimation of Expected Step Quality

To estimate the expectation of step quality, we average the step quality of all $B \times E$ candidates at each time step. For each candidate, we adopt the Monte Carlo estimation method from [17]. Specifically, starting from each candidate, we generate $N$ subsequent trajectories (we set $N = 4$ in our experiments) and approximate the step quality by the proportion of trajectories that reach the correct answer. The experiments in [17] also demonstrate that using a small $N$ (5 in their experiments) does not introduce significant bias.

### B.3  Step Segmentation

In general, step segmentation can be based on symbolic cues (e.g., explicit "Step" annotations or line breaks), model confidence[19], or token-based heuristics. Given that `DeepSeek-R1-Distill-Qwen` models do not yield outputs with clearly demarcated step boundaries, we adopt a hybrid segmentation strategy that integrates symbolic markers and token-length priors.

Specifically, we treat certain tokens—such as line breaks (\n) or step-indicative phrases—as potential split points, and further apply this rule approximately every 2000 tokens to prevent overly fragmented segmentation in long outputs. This approach yields around 10–20 steps per output in practice and works well empirically.

## C  Additional Experiments and anlysis

### C.1  Hyperparameter affects the aggressiveness of early stopping.

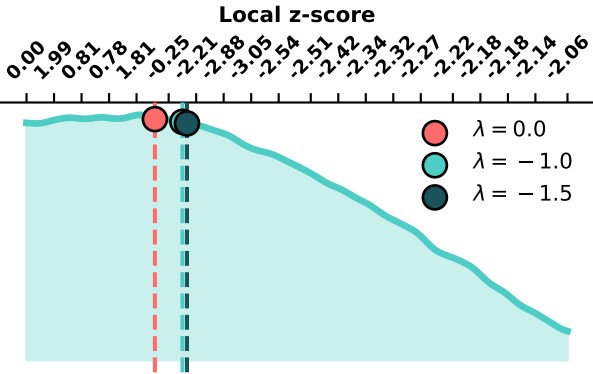

Figure 7: Smaller values of $\lambda$ result in earlier termination points.

We illustrate the influence of the parameter $\lambda$ on the stopping position using the concave function in Figure 7 as an example. Specifically, we compute the local Z-score at each discrete point (i.e., by performing Z-normalization on the values from the starting point up to the current point). We then apply three different values of $\lambda$. As $\lambda$ decreases, the stopping point moves later (or is delayed); conversely, a larger $\lambda$ value leads to an earlier stopping position (or advances).

## C.2 Empirical Evidence for Assumption 3.2

**Assumption 3.2** states that for a *reasoning-capable* policy, the re-ranking ability of the PRM gradually declines as Beam Search progresses—it becomes increasingly difficult to select the optimal next step from the candidate set.

To empirically validate this, we consider **DeepSeek-R1-Distill-7B** as a reasoning-capable policy, as evidenced by its performance on the AMC2023 benchmark, where it achieves 90% accuracy without relying on any test-time methods (see Table 1). We apply the **Hard PRM** to guide Beam Search and evaluate PRM's re-ranking ability at each step by checking whether it can successfully identify the optimal next step from the $B \times E$ candidate actions.

The optimal next step is determined using the Monte Carlo estimation of step quality described in Appendix B.2.

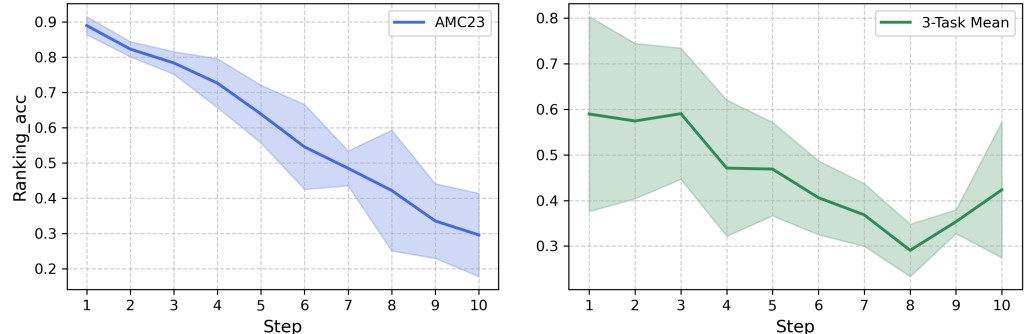

Figure 8: Re-ranking accuracy of Hard PRM on AMC2023

Figure 9: Re-ranking accuracy of Hard PRM on three benchmarks

We follow the above methodology to measure the re-ranking accuracy of PRM under different Beam Search configurations (Beam ranging from 4 to 64 with Expansion fixed to 2, same as in Section 3.1). The results are shown in Figure 8 and Figure 9.

Figure 8 shows the accuracy measured exclusively on the AMC2023 benchmark, where the policy exhibits stronger *reasoning* capabilities, thereby more closely aligning with the assumptions underlying our hypothesis. Figure 9 shows the average re-ranking accuracy across three benchmarks: AMC2023, AIME2024, and AIME2025.

From the results, it is evident that the PRM's re-ranking ability declines as Beam Search progresses, and this decline is quite pronounced. On the AMC2023 benchmark, the decrease is even more striking—almost linear. This provides strong support for the conjecture in Claim 3.7, indicating that PRM's performance deteriorates progressively as Beam Search advances. Consequently, the expectation of step quality exhibits concavity, which in turn gives rise to the step quality degradation discussed in Section 3.3.

# D  Related Works (prior work on early stopping)

Previous studies have explored early stopping algorithms for beam search [20, 21]. These studies focus on translation-style generation tasks, where hypotheses are ranked by likelihood and length penalties, and have made important contributions to decoding strategies. In contrast, our work focuses on PRM-guided inference under the Long CoT model setting. We introduce a novel z-score-based termination criterion motivated by empirical observations of PRM dynamics, which improves stopping reliability and reasoning performance.

