# OpenReview forum: "Reasoning Is Not a Race: When Stopping Early Beats Going Deeper"
_NeurIPS.cc/2025/Conference — NeurIPS 2025 poster_

### Official Review · Reviewer_NNk9 · 2025-06-19

**Clarity:** 2
**Significance:** 3
**Originality:** 3
**Rating:** 4
**Confidence:** 4

**Summary:**

First, this study has empirically found that the PRM quality during beam search degrades as step increases, making unimodal curve.
Theoretical analysis also support the claim by deriving that the second order of the curve becomes minus value if we assume that the PRM’s re-ranking capacity gradually declines during beam search.

Second, it has proposed a method of early stopping beam search by setting the stopping trigger as the z-normalized PRM reward falling below a threshold. Experiments have demonstrated that the proposed method consistently outperform or competitive as the baseline methods (normal PRM beam search and self-consistency method), insisting the effectiveness of the proposed method.

**Questions:**

I would suggest that the authors update manuscripts to increase readability and clarity.
- Figure1: This overview lacks lots of important information. First, there is no separate explanation for each of (a) and (b) in the caption. Second, The figure itself does not illustrate what to do after the early stopping (so readers cannot grasp the overall concept of the proposal at first glance). Third, there is no explanation in the caption/figure about what is "locally normalized z-score" and why it degrades as the steps increases. The explanation should not be detailed, but it should be simple but to the point.
- "Step quality" is an important concept in this paper, but the detailed calculation approach was moved to Appendix B.2. It is better to describe it in Main page for readability.
- Step 4 of the proposed method "stop Beam Search early and directly decode from the candidates": what kind of decoding strategy is used?

**Ethical Concerns:**

["NO or VERY MINOR ethics concerns only"]

**Final Justification:**

I acknowledge that some of my concerns are resolved by the reply, but the core idea of the proposed method is heuristic and has less grounded on the finding theory of the degradation pattern of PRM, which may cause readers feel a gap between them (In a way, I feel like reading separated two papers though both of the sections are of good quality).

However, after second consideration, my remaining concerns are trivial (i.e., if degradation occurs at the very beginning a unimodal pattern will not emerge, Local z-score is unstable in early steps, missing manually annotated PRM baseline), so I have raised my score from borderline reject to borderline accept.

**Limitations:**

Yes.

**Paper Formatting Concerns:**

No.

**Quality:**

3

**Strengths And Weaknesses:**

Strengths:
- This study carefully observe the deficiency of PRM-based beam search and found that the cause of inferior performance was the consistent degradation of PRM quality as step increases. This finding is beneficial for the community.
- The proposed method has shown to be effective across different size of models and multiple benchmarks, outperforming or competitive as major baselines.

Weaknesses:
- The theoretical analysis does not always support the claim that there exists unimodal peak. For instance, if the degradation happens at the begging step, the curve consistently decreases as step increases and there is no peak according to the theoretical consequence that second-order difference of the expected step quality is minus.
- The proposed method is heuristic and not exciting, just early stopping the beam search.
- Step 3 of the proposed method "Compute the local z-score using the sequence of {x1, x2, ..., xt}": In the early phase of the step, the size of the sequence is so small, so the normalized z-score becomes unstable.
- Ideally, the baseline method should include manually annotated PRM approach to indicate an upper-bound of PRM.
- I think there are some readability and clarity issues. Please see "Questions".

---

> ### Author Rebuttal · Authors · 2025-07-31
>
> **We sincerely thank the reviewer for the thoughtful and constructive comments. Below, we address each of your points in detail.**
>
> ---
>
> ### Weaknesses
>
> ---
>
> >#### 1. **“Theoretical analysis does not always support the claim that there exists unimodal peak...”**
>
> We thank the reviewer for this insightful observation. We agree that if degradation occurs at the very beginning, a unimodal pattern will not emerge. As clarified in our paper, step quality degradation can take two forms: unimodal and monotonically decreasing (see lines 135–136). Our theoretical analysis explicitly covers and explains both cases (see lines 205–212).
>
> ---
>
> >#### 2. **“The proposed method is heuristic and not exciting...”**
>
> We thank the reviewer for raising this point. However, we respectfully disagree that our method lacks novelty. Our contributions are as follows:
> 1. We investigated the step quality dynamics of PRM-guided Beam Search under the long CoT model setting and empirically demonstrated the existence of the step quality degradation phenomenon (to the best of our knowledge, we are the first to do so).
> 2. We provided a theoretical explanation for why this degradation occurs.
> 3. Based on both theoretical and empirical analysis, we proposed an adaptive early stopping method, ZGES, which is simple to implement yet achieves strong performance across multiple tasks and models. Moreover, it can be applied to more general search methods such as look-ahead search and outperforms baselines (see our response to Reviewer egya, Weakness 1). Therefore, our method is not limited to Beam Search but can broadly benefit search-based test-time scaling methods for LLM reasoning tasks.
>
> ---
>
> >#### 3. **“Local z-score is unstable in early steps...”**
>
> We appreciate your insight and the question raised. We acknowledge that statistical instability exists when the sample size is small. However, in our implementation, we apply a technique that sets a minimum early stopping step (e.g., set to 3), meaning ZGES will not perform early stopping before step 3, which effectively mitigates the instability issue. We will include this detail in the revised manuscript.
>
> ---
>
> >#### 4. **“Missing manually annotated PRM baseline...”**
>
> We appreciate the reviewer’s suggestion and agree that manually annotated data can provide a higher-quality PRM as an upper bound baseline.
>
> However, as noted in our response to Weakness 2, research on PRM-guided test-time scaling for long CoT models is limited, and no publicly available manually annotated PRM data exists for this setting.
>
> Due to the high cost of manual annotation, most methods rely on automatically annotated data, which better suits practical efficiency needs.
>
> We will clarify this limitation in the revised manuscript and highlight developing high-quality manual PRMs as future work to improve performance and baseline comparisons.
>
>
> ---
>
> ### Questions and Clarity Issues
>
> ---
>
> >#### Q1: *Figure 1 lacks clarity*
>
> We thank the reviewer for these constructive comments. We agree that Figure 1 can be improved for clarity. In the revision, we will:
> 1. Provide separate, concise explanations for subfigures (a) and (b) in the caption.
> 2. Update the figure to illustrate the procedure after early stopping so that the overall method can be understood at a glance.
> 3. Add a brief explanation in the caption/figure about what the locally normalized z-score is and why it tends to decrease as the step count increases.
>
> We believe these changes will make Figure 1 more self-contained and easier for readers to grasp the main idea.
>
> ---
>
> >#### Q2: *Step quality definition relegated to Appendix*
>
> We thank the reviewer for pointing this out. We agree that "step quality" is a key concept in our work and that describing its calculation in the main text would improve readability. In the revision, we will move a concise version of the calculation from Appendix B.2 to the main paper, while keeping the full details in the appendix for completeness.
>
> ---
>
> >#### Q3: *Step 4 decoding strategy*
>
> We thank the reviewer for the question. In Step 4, after early stopping, we decode from the remaining candidates using **nucleus sampling (top‑p)** with `p = 0.95`. The temperature is set to `0.6` for the 1.5B model and `1.0` for the 7B model.

---

> > ### Comment · Reviewer_NNk9 · 2025-08-04
> >
> > Thank you for the author’s reply.
> >
> > I acknowledge that some of my concerns are resolved by the reply, but the core idea of the proposed method is heuristic and has less grounded on the finding theory of the degradation pattern of PRM, which may cause readers feel a gap between them (In a way, I feel like reading separated two papers though both of the sections are of good quality).
> >
> > In addition, I hope the authors refine the manuscript in the next revised version to improve the readability as pointed out.

---

### Official Review · Reviewer_WZXW · 2025-06-23

**Clarity:** 3
**Significance:** 3
**Originality:** 3
**Rating:** 5
**Confidence:** 3

**Summary:**

This paper studies the performance degeneration of Beam Search as the decoding step grows in searching for high-quality long CoT with a Process Reward Model (PRM). The authors define the step quality as the value of the state and estimate it using Monte Carlo simulation. Empirically, they observed that the step quality decreases significantly as the decoding step grows. They also theoretically show that the expected step quality is concave in the discrete sense.  To solve the degradation of step quality, the authors propose to use PRM scores as a proxy (based on a correlation analysis) and early terminate beam search when the z-score below a threshold. The merits of the proposed method are empirically verified.

**Questions:**

See my comments in Strengths And Weaknesses section

**Ethical Concerns:**

["NO or VERY MINOR ethics concerns only"]

**Final Justification:**

I am satisfied with authors' rebuttal. Therefore I maintained my score

**Limitations:**

yes

**Paper Formatting Concerns:**

N.A.

**Quality:**

3

**Strengths And Weaknesses:**

I think the authors do a very well job in presenting their core ideas. The problem to be solved is important, the analysis are intuitive and the claims are well supported, either empirically or theoretically.

I have only some minor concerns regarding the paper.
 - First, although your method for early stopping is intuitive, it seems to have nothing to do with the theoretical results you have derived in Section 3.3. Can you shed some lights on how these results (especially Proposition 3.8) could be used in developing early stopping/pruning algorithms for BS or more generally, tree-search-based test-time scaling methods?
 - Second, note that the hyperparameter $\lambda$ can be sensitive to the PRMs. One might need to tune case-by-case for different scenarios.
 - Third, I suggest the authors adding the results of token usages or the number of PRM calls, otherwise line 279-280 is not very well supported.

Some minor presentation issues

 - line 23: O1 -> o1
 - line 46: insert a space after word "concavity"
 - line 85: it seems that $r_h$ should map to {$0,1$}, not $[0,1]$
 - Figure 5: the negative sign is too thin and almost invisible
 - How $\lambda$ is set in your main results (Figure 4)?

---

> ### Author Rebuttal · Authors · 2025-07-30
>
> **We sincerely thank the reviewer for their thoughtful comments, positive feedback, and constructive suggestions. Below we address each point in detail.**
>
> ---
>
> ### **Comment 1** Connection Between Theoretical Results and Early Stopping
>
> We thank the reviewer for this insightful question. In Section 3.2, we observe the phenomenon of step quality degradation — namely, the expected step quality typically exhibits either a unimodal shape or a monotonically decreasing pattern. This empirical observation indicates that early stopping can be beneficial for the tasks considered in our experiments.
>
> In Section 3.3, we provide a theoretical explanation for this phenomenon. Specifically, `Proposition 3.8` establishes the concavity of step quality. As discussed in lines 209–212, both common forms of concave functions exhibit a peak, which offers theoretical support for early stopping. Combining this empirical evidence with our theoretical analysis naturally leads to our heuristic `ZGES` method.
>
> Regarding your question on extending the theoretical insight beyond beam search to more general `tree-search-based test-time scaling methods`, we fully agree that this is a valuable direction. We are excited to report that we have already applied `ZGES` to a generalized form of beam search — `look-ahead search` (see our response to Reviewer egya, Weakness 1) — and found that `ZGES` consistently outperforms the standard look-ahead search. This suggests that the concavity of step quality revealed by our theory may be a general property, and we plan to further explore its theoretical implications for other forms of tree-search-based test-time scaling methods in future work.
>
>
> ---
>
> ### **Comment 2** Sensitivity of λ
>
> We thank the reviewer for astutely pointing this out — we agree with your assessment. Indeed, hyperparameter tuning is a common challenge across most methods involving tunable parameters. We were aware of the potential sensitivity of our hyperparameter to PRMs, as you noted, and accordingly conducted a **sensitivity analysis** in Section 5.1 of the paper.
>
> Specifically, we evaluated our method on both `1.5B` and `7B` models using different values of `λ` (-0.4, -0.6, -0.8, -1.0), and found that performance differences were **not significant** across these values (see Figure 5). This suggests that for both tested models, our method is relatively robust to `λ` within this range.
>
> Altogether, these experiments provide an initial empirical analysis of ZGES’s sensitivity to model scale, and suggest that the method exhibits **a degree of robustness**. We agree that developing **hyperparameter-free early stopping mechanisms** is an interesting and valuable future direction, and we leave this for future work.
>
> ---
>
> ### **Comment 3** Token Usage and PRM Calls
>
> We thank the reviewer for the helpful suggestion, and we fully agree that including statistics on PRM call counts and token usage can strengthen the empirical justification of our claims in lines 279–280, thereby providing further evidence for the advantages of our method.
>
> Following your suggestion, we conducted additional analysis to report these statistics. The results are summarized as follows:
> | Model | Method   | AMC23 PRM Call | AMC23 Tokens Ratio | AIME24 PRM Call | AIME24 Tokens Ratio | AIME25 PRM Call | AIME25 Tokens Ratio |
> |:-----------:|:-----------:|:---------:|:------------:|:----------:|:-------------:|:----------:|:-------------:|
> | 1.5B| **ZGES (-0.4)** |   **3.08** ↓61%    |   **0.753**     |    **3.27** ↓66%    |   **0.929**     |    **3.33** ↓65%   |    **0.926**     |
> | 1.5B| **ZGES (-0.8)** |   **3.18** ↓60%    |   **0.763**     |    **3.42** ↓64%    |    **0.956**    |    **3.48**  ↓63%  |    **0.942**     |
> | 1.5B| Beam Search  |   7.91    |   ---     |    9.55    |    ---     |    9.53    |    ---     |
> | 7B| **ZGES (-0.4)**   |   **2.97** ↓56%    |   **0.675**     |    **3.31** ↓63%   |    **0.839**     |    **3.33** ↓62%    |    **0.905**     |
> | 7B| **ZGES (-0.8)**   |   **3.17** ↓53%    |   **0.692**    |    **3.54**  ↓61%  |    **0.856**     |    **3.58**  ↓59%  |    **0.911**     |
> | 7B| Beam Search    |   6.69    |   ---     |    9.00    |   ---     |    8.73    |    ---     |
>
> **Note:**
> 1. `1.5B` represents the R1-Distill-Qwen-1.5B model, and `7B` represents R1-Distill-Qwen-7B.
> 2. `ZGES(x)` denotes the ZGES method (applied to Beam Search) with λ set to x, while Beam Search refers to the standard Beam Search method.
> 3. The `PRM Call` column shows the average number of PRM calls, and `Tokens Ratio` represents the average number of tokens generated by ZGES relative to Beam Search.
>
>
> We can observe three interesting points:
>
> - Firstly, the experimental results strongly support the statement in lines 279-280 of our paper, namely that compared to the standard PRM-Guided Beam Search, the ZGES method can significantly reduce the number of PRM calls (by more than `50%`) without sacrificing performance, and even achieve better results (according to the experimental results in Section 4.3 of the paper).
>
> - Secondly, for the 1.5B or 7B models, we find that the smaller the value of `λ`, the more PRM calls are made. This is consistent with the theoretical prediction in Appendix C.1, where a smaller `λ` causes the stopping point to be delayed, resulting in more PRM calls.
>
> - Finally, we observe that from amc23 to aime25, as the difficulty of the problems increases, the number of PRM calls (for the ZGES method) also increases. We hypothesize that when the problem difficulty is higher, increasing exploration is beneficial.
>
> ---
>
> ### **Comment 4** Minor Presentation Issues
>
> We appreciate your attention to detail and will address the following corrections in the revised version:
>
> - Line 23: "O1" → "o1"
> - Line 46: Add space after "concavity"
> - Line 85: Change the codomain of $ r _ h $ to  {0,1}
> - Figure 5: Fix the thin negative sign for better visibility
> - Clarify how $ \lambda $ is set in Figure 4 (we will add this in the caption or appendix)
>
> **Regarding the final point**, we would like to clarify that in our main results (Figure 4), we chose a specific value of λ for each model: `λ` = -0.6 for the 1.5B model and `λ` = 0 for the 7B model. However, we would like to emphasize that, as we noted in our response to Comment 2, the performance is generally not highly sensitive to the choice of `λ`. That is, different `λ` values do not lead to significant differences in the results across models.
>
> ---
> ### **Once Again**
>
> We thank the reviewer for their detailed and helpful review. We will incorporate all the above revisions to improve the clarity and rigor of our paper.

---

> > ### Comment · Reviewer_WZXW · 2025-08-05
> >
> > Thanks for response. Generally, I am satisfied with authors' rebuttal. However I think "when the problem difficulty is higher, increasing exploration is beneficial" is somewhat misleading. Actually, the behavior of ZGES has nothing to do with either "increasing exploration" or "problem difficulty". Only factor that affects ZGES is the dynamics of PRM scores (given fixed lambda). Just curious, can you say something about PRM dynamics across different benchmarks?

---

> > > ### Author Response · Authors · 2025-08-08
> > >
> > > We thank the reviewer for pointing out the imprecision in our previous statement. The observed increase in PRM calls and token usage is not due to ZGES explicitly increasing exploration based on task difficulty, but rather is a **natural consequence of how PRM dynamics evolve** in more challenging environments.
> > >
> > > In response to the reviewer's question about PRM dynamics across different benchmarks, our paper's `Figure 3` and `Table 2` already show an important empirical phenomenon: across various datasets and model scales, there is a **strong linear correlation between PRM reward and step quality**. This indicates that the dynamics of PRM scores effectively reflect the evolution of step quality during decoding, and this trend remains consistent across all benchmarks.
> > >
> > > We thank the reviewer again for their careful observation and insightful question.

---

> > > > ### Comment · Reviewer_WZXW · 2025-08-08
> > > >
> > > > Thanks for your response.

---

### Official Review · Reviewer_LAc4 · 2025-06-30

**Clarity:** 2
**Significance:** 3
**Originality:** 3
**Rating:** 5
**Confidence:** 2

**Summary:**

The work provides empirical evidence that Progress Reward Models (PRM) are ineffective for long Chain-of-Thought reasoning tasks and offers theoretical intuition and empirical analysis to provide the following explanation: The PRMs ability to correctly estimate the probability that the LLM finds the correct answer from a given token first increases and then decreases during the search process. An empirical analysis shows that this probability is highly correlated with the PRM score. This insight is leveraged to define an early stopping criterion for the search. The stopping criterion is based on the z-scores of the PRM scores: As soon as it drops below a certain threshold the search is stopped. A comparison with majority@N and standard PRM-guided beam search shows that the method achieves higher accuracy on reasoning tasks.

**Questions:**

* In the experiment in Section 4.3: How did you define the stopping criterion for the baselines?
* What is $\tau$ in Assumption 3.1?
* How is the correctness of a state defined?
* In Section 3.1, Equation (1): $\pi$ is mentioned to be the LLM policy. What is the LLM policy? Is it PRM-guided beam search in this context?
* If I understood it correctly, I would have assumed that the pattern in Figure 2 looks like the pattern in Figure (b). But in Figure 2, the step quality doesn't have this peak. Can you elaborate on this?
* The theoretical analysis assumes a beam size of 1 (see line 165). This essentially reduces it to greedy search. To me, this seems like quite a strong restriction. Can you elaborate on this?

**Ethical Concerns:**

["NO or VERY MINOR ethics concerns only"]

**Final Justification:**

My concerns were addressed with the rebuttal, therefore I increased my score.

**Limitations:**

yes

**Paper Formatting Concerns:**

no formatting concers

**Quality:**

3

**Strengths And Weaknesses:**

**Strengths:**
* The paper provides a theoretical explanation for the empirical phenomenon that the performance of PRM-guided beam search drops in the **long** chain of reasoning regime
* The idea to use z-scores to define an early-stopping criterion is original as far as I know
* The work is significant as early-stopping helps to reduce computational costs/increases the performance in LLM-reasoning tasks
* The paper provides an ablation study for the involved hyper parameters (stopping threshold, beam size). The results show that the method is relatively robust towards these choices.

**Weaknesses:**
* Clarity: there are some parts of the submission that are unclear to me. I am open to increasing my score if these issues can be clarified and resolved. I listed my questions below.
* Related work: The related work section does not touch upon other pieces of work on early-stopping (a.k.a length normalization) for beam search such as:
  > Google’s Neural Machine Translation System: Bridging the Gap between Human and Machine Translation (Wu et al., 2016)

  > When to finish? optimal beam search for neural text generation (modulo beam size). (Huang et al., 2017)

  > Breaking the beam search curse: A study of (re-)scoring methods and stopping criteria for neural machine translation.

**Minor comments:**
* The legend in Figure 6 covers most of the plot.
* lines 236-244 contain a proof for a theorem/proposition/statement that is never explicitly stated. It is stated informally "... we demonstrate the the t-scores of the two variables remain consistent throughout the search process.", but stating it explicitly would make it easier to refer to it/cite it in future work ect.
* "concave in the discrete sense" $\rightarrow$ isn't the discrete analogue to concavity simply submodularity? Is there are reason why you don't use this term?
* For me, it would have been helpful if there was a definition of the "Majority@N" strategy included, in particular because there is no reference to a publication that introduces this method. I was not familiar with it.

---

> ### Author Rebuttal · Authors · 2025-07-30
>
> **We thank the reviewer for the thoughtful feedback and constructive suggestions. We are encouraged by the recognition of our contributions. Below, we address each of the reviewer’s concerns in detail.**
>
> ---
>
> ### Clarity Issues
>
> >**Comment:**
> *There are some parts of the submission that are unclear to me. I am open to increasing my score if these issues can be clarified and resolved.*
>
> **Response:**
> Thank you for your openness. Specific clarifications are addressed in responses to your detailed questions below.
>
> ---
>
> ### Related Work on Early Stopping
>
> >**Comment:**
> *The related work section does not mention prior work on early stopping / length normalization for beam search such as Wu et al. (2016), Huang et al. (2017), and others.*
>
> **Response:**
> We thank the reviewer for highlighting the relevant work on early stopping in beam search. These studies focus on translation-style generation tasks, where hypotheses are ranked by likelihood and length penalties. They have made important contributions to decoding strategies.
>
> In contrast, our work focuses on *PRM-guided inference* under the *Long CoT* model setting. We introduce a novel **z-score-based termination criterion** motivated by empirical observations of PRM dynamics, which improves stopping reliability and reasoning performance.
>
> We will add a discussion of these works in the *Related Work* section.
>
>
> ---
>
> ### Minor Comments
>
> >**Comment:**
> *Figure 6 legend covers most of the plot.*
>
> **Response:**
> Thank you for pointing that out. We will correct the occlusion of the Figure 6 legend in the revised version of the paper.
>
> >**Comment:**
> *Lines 236–244 include a proof but no explicit theorem statement.*
>
> **Response:**
> We thank the reviewer for the insightful comment. You're right — our earlier statement lacked formality. In the revision, we will include a lemma to formally state the z-score consistency.
>
>
> >**Comment:**
> *"Concave in the discrete sense" – why not use "submodular"?*
>
> **Response:**
> We sincerely thank the reviewer for this thoughtful comment. We agree that **submodularity** is the standard discrete analogue to concavity, and we appreciate the suggestion.
>
> In our work, the step quality function is defined over a one-dimensional discrete time index$t \in \{1, 2, \ldots, T\}$
> , rather than subsets of a ground set. We intentionally use the phrase “concave in the discrete sense” to provide an intuitive and accessible description of the diminishing trend in step quality over time. Our goal is to make this behavior immediately understandable to a broader audience, especially readers less familiar with the formalism of submodular set functions.
>
> Specifically, we refer to functions \( f \) defined on integers \( t \) satisfying the **discrete concavity condition**:
>
> $$
> f(t+1) - f(t) \leq f(t) - f(t-1)
> $$
>
> which parallels the classical second-derivative criterion for concavity in continuous settings.
>
> That said, we fully acknowledge the reviewer’s point, and in the revised version of the paper, we will include a clarification on the connection between our definition and submodularity, to ensure greater precision and rigor.
>
>
> >**Comment:**
> *Define Majority@N clearly – no citation provided.*
>
> **Response:**
> Thank you for the helpful suggestion. We would like to clarify the **Majority@N** strategy (also known as *self-consistency*):
> Formally, let the model generate a set of answers $\{a_1, a_2, \ldots, a_N\}$. The final prediction is:
>
> $$
> \hat{a} = \arg\max_{a \in \{a_1, \ldots, a_N\}} \text{Count}(a)
> $$
>
> i.e., the most frequent answer among the N samples.
> To enhance the clarity and readability of the paper, we will add a formal definition of Majority@N in the revised version.
>
> ---
>
> ### Responses to Specific Questions
>
> >**Q1: In the experiment in Section 4.3: How did you define the stopping criterion for the baselines?**
>
> **A:** Thank you for the reviewer’s question. As stated in Section 4.3, the baselines include Majority@N and standard PRM-Guided Beam Search, neither of which employs early stopping.
>
>
> ---
>
> >**Q2: What is $\tau$ in Assumption 3.1?**
>
> **A:** Thank you very much for your valuable question.
> In our setting, $\tau$ denotes a complete reasoning trajectory generated by the model when solving a given problem. It consists of a sequence of reasoning steps, i.e.,
>
> $$
> \tau = \{a_1, a_2, \ldots, a_T\},
> $$
>
> where $a_T$ (the final step) typically contains the model's predicted answer.
>
> As we mention in Section 3.3, we would like to emphasize that under Assumption 3.1:
>
> * A trajectory denoted as $\tau$ refers to a **correct** trajectory — that is, one where the final answer in $a_T$ is correct.
> * A trajectory denoted as $\bar{\tau}$ refers to an **incorrect** trajectory — that is, one where the final answer in $a_T$ is incorrect.
>
> We will revise the paper to ensure this definition is clearly and consistently stated.
>
> ---
>
> >**Q3: How is the correctness of a state defined?**
>
> **A:**
>
> Thank you for the reviewer’s insightful question. In Assumption 3.1, we follow the convention in [2], where state correctness is not explicitly defined per state but is implicitly captured by the quality of future outcomes. We will make this connection explicit in the revision.
>
> [2] *PROCESS REWARD MODEL WITH Q-VALUE RANKINGS, ICLR 2025.*
>
> ---
>
> >**Q4: What is $\pi$ in Equation (1)?  Is it PRM-guided beam search in this context?**
>
> **A:**
> Thank you for the question. In Equation (1), the *LLM policy* refers to the reasoning-capable LLM (in our setting, the Long CoT Model such as DeepSeek-R1-Distill-Qwen). We refer to it as a *policy* to align with our formulation in Section 2, where we model LLM reasoning as a Markov Decision Process (MDP).
>
> We would like to clarify that the definition of *LLM policy* is independent of PRM-guided beam search. In other words, the *step quality* defined in Equation (1) is not based on the PRM-guided beam search procedure. Instead, it is defined under the context of letting the LLM directly generate.
>
>
> ---
>
> >**Q5: Why doesn’t Figure 2 show a peak in step quality like Figure 2b?**
>
> **A:** We sincerely thank the reviewer for the question. Your understanding is absolutely correct — the empirical curves in Figure 2 are indeed expected to follow a similar pattern as the illustrative curve shown in Figure 1(b).
>
> We would like to clarify that there do exist peaks in the curves of Figure 2, and these peaks typically occur at the second decoding step (i.e., around the 4k mark on the x-axis). However, these peaks are sometimes subtle and thus may be difficult to visually observe. That said, some curves do exhibit more prominent peaks, such as the statistics of the 1.5B model on the AIME24 dataset.
>
> ---
>
> >**Q6: Why is beam size = 1 assumed in theoretical analysis? Isn’t that greedy search?**
>
> **A:**
> We thank the reviewer for the insightful question. You are absolutely right — setting the beam size to 1 is indeed equivalent to greedy search. We would like to clarify that this simplification is made purely for the ease of mathematical presentation and does not affect the overall problem formulation.
>
> If the beam size is set to $N$, then in Equation (2), when computing the expectation of step quality, the expression
>
>
> $$ P^{\pi} _ {\text{BS}}(s _ t \mid s _ 0) V^{\pi}(s _ t) + P^{\pi} _ {\text{BS}}(\bar{s} _ t \mid s _ 0) V^{\pi}(\bar{s} _ t) $$
>
>
> should be replaced with a sum over $N$ terms, i.e.,
>
> $$
> \frac{1}{N} \sum _{i=1}^{N} \left( P^{\pi} _ {\text{BS}}(s _ t^i \mid s _ 0) V^{\pi}(s _ t^i) + P^{\pi} _ {\text{BS}}(\bar{s} _ t^i \mid s _ 0) V^{\pi}(\bar{s} _ t^i) \right)
> $$
>
> where $s_t^i$ denotes the $i$-th beam candidate. The rest of the analysis and conclusions remain consistent under this more general form. We will annotate this point explicitly in the revised version for improved clarity and completeness.
>
> ---
>
> ### Final Remarks
>
> We thank the reviewer again for the constructive review. We believe the revised paper better clarifies our contributions, improves alignment with related work, and resolves the open questions. We hope our responses and revisions address your concerns and demonstrate the significance of our findings.

---

> > ### Comment · Reviewer_LAc4 · 2025-08-05
> >
> > **Thank you very much for the explanations! Most of my questions have been answered sufficiently.**
> >
> > I am still not entirely sure whether I understand the connection with early termination by length normalization for beam search as described by Wu et al. (2016), Huang et al. (2017), and others. Is there anything that prevents the application of a length normalization criterion in the PRM-driven inference environment or is it just that it hasn't been studied before?

---

> > > ### Author Response · Authors · 2025-08-08
> > >
> > > Thank you for your question, which prompts us to further clarify the connection between our work and prior approaches to early stopping in beam search.
> > >
> > > ---
> > >
> > > >### **Clarification on Length Normalization**
> > >
> > > First, it is important to distinguish the purpose of length penalties (or normalization/rewards) in previous work. In models for tasks like machine translation (e.g., the work by Wu et al., 2016), these methods were primarily a `rescoring technique`, not an early stopping strategy. They were designed to prevent the model from generating overly short sentences, a common issue in those tasks.
> > >
> > > In our `PRM-driven inference environment`, the problem of overly short outputs is not a significant concern, especially in long CoT model settings. Therefore, introducing a length normalization mechanism for this purpose is not necessary in our context.
> > >
> > > ---
> > >
> > > >### **Fundamental Differences in Problem Formulation**
> > >
> > > Secondly, we'd like to elaborate on the core differences between the PRM-driven inference environment and the issues faced by previous translation work. We believe there are two key distinctions:
> > >
> > > 1.  **Scoring Mechanism:** In prior work, candidate translations were scored based on the product of model output probabilities (with some rescoring methods like length rewards). In contrast, the PRM-driven environment relies on a `step-wise scoring` from a Process Reward Model (PRM). This means the score for a step is given by the PRM at that moment, not by accumulating scores from previous steps. This fundamental difference makes prior early stopping criteria, such as the optimal stopping rule mentioned in Wu et al. (2016) inapplicable in principle.
> > >
> > > 2.  **Answer Selection:** For tasks like translation, the final answer is a single best candidate from a set of hypotheses. For our reasoning tasks (e.g., math problems), there is a definitive correct answer. This allows us to use an aggregation method like **Majority@N**, which is not feasible for translation. This difference in final answer selection also means that previous beam search stopping methods are not directly applicable.
> > >
> > > ---
> > >
> > > In summary, the distinct scoring mechanism and task formulation necessitate a different approach to early stopping. While our method is tailored to the unique properties of PRM-guided reasoning, we agree that there are valuable insights to be gained from previous work. For instance, if we were to discard the PRM and rely on model confidence (output probabilities) for scoring, we might be able to draw a closer parallel and adapt some of their ideas for improved performance. We believe exploring such connections would be an interesting direction for future research.
> > >
> > > We hope this response is helpful in clarifying the distinctions, and we thank you again for your insightful question!

---

> > > > ### Comment · Reviewer_LAc4 · 2025-08-08
> > > >
> > > > Thank you for the more detailed explanation. The rebuttal addressed my concerns and I update my score accordingly.

---

### Official Review · Reviewer_egya · 2025-07-01

**Clarity:** 3
**Significance:** 2
**Originality:** 3
**Rating:** 3
**Confidence:** 4

**Summary:**

This paper investigates the effectiveness of Process Reward Models (PRMs) in guiding test-time search for Long Chain-of-Thought (Long CoT) reasoning. The authors find that standard PRM-guided beam search does not consistently outperform PRM-free approaches like Majority@N. Through empirical and theoretical analysis, they observe a phenomenon of step quality degradation during beam search and propose Z-Score Guided Early Stopping (ZGES) as a remedy. ZGES monitors local trends in PRM reward signals and adaptively halts beam expansion when reasoning quality peaks, thereby reducing unnecessary computation and mitigating the diminishing returns of deeper search. The method is evaluated on Long CoT math benchmarks and shows improved accuracy and efficiency compared to conventional beam search methods.

**Questions:**

See above weakness

**Ethical Concerns:**

["NO or VERY MINOR ethics concerns only"]

**Final Justification:**

While the rebuttal clarified some minor points, the two core concerns remain insufficiently addressed. First, the reply lacks theoretical justification or detailed analysis to explain why the proposed method shows such a large performance gap between long-CoT and short-CoT settings, or with AlphaGo-style algorithms, and it does not address my request for experiments on more existing PRMs. Second, although the authors acknowledged a critical limitation, no preliminary attempt was made to run even a lightweight evaluation on the test set, which would incur minimal computational cost. These two points are central to both the method and its claimed effectiveness, and remain unresolved after the rebuttal.

That said, I acknowledge that the paper presents some new perspectives, so I would understand if it is ultimately accepted, but my final score remains cautious given the unresolved core issues.

**Limitations:**

yes

**Quality:**

3

**Strengths And Weaknesses:**

Strengths

1. The paper targets an important and practical challenge in reasoning with LLMs: improving test-time search in Long CoT settings using PRMs.

2. The authors identify and explain the degradation of step quality during beam search with theoretical justification (concavity of step quality trajectory).

3. ZGES is a simple yet effective strategy that improves performance while reducing compute, and is well supported by experiments.

Weaknesses

1. The findings partially contradict prior conclusions in PRM literature (e.g., Scaling LLM Test-Time Compute Optimally, AlphaZero-style lookahead), which have shown benefits from deeper reward-guided exploration. The paper would be strengthened by applying ZGES to a wider range of PRMs (e.g., value estimation PRMs trained with lookahead) to ensure its generality and reconcile these differing observations.

2. The training process of the PRM used in this study is under-explained. There is little analysis of PRM accuracy, which is critical given that poorly trained PRMs may introduce unreliable reward signals. Related benchmarks like PRMBench could help validate the PRM’s fidelity.

3. All experiments are limited to math datasets. Testing on other domains such as scientific QA (e.g., GPQA) would be valuable to demonstrate the generalization of ZGES beyond math reasoning.

4. Since the Long CoT outputs from models like DeepSeek-R1 are not cleanly step-separated, the process of segmenting steps is crucial for PRM-guided reasoning, yet the paper provides no technical detail or justification for how step boundaries are identified.

5. Minor writing issues exist, e.g., line 87 contains a typo: “reasonoing”.

---

> ### Author Rebuttal · Authors · 2025-07-29
>
> **We thank the reviewer for the thoughtful feedback and valuable suggestions. We address each point below and believe our clarifications and additions help strengthen the paper.**
>
> ---
>
>
> >**Weakness 1. Contradiction with prior PRM literature & generality of ZGES**
>
> We thank the reviewer for highlighting the apparent discrepancy between our findings and prior PRM-related work, such as *Scaling LLM Test-Time Compute Optimally can be More Effective than Scaling Model Parameters*. That work demonstrates that reward-guided exploration methods at test time—such as beam search and lookahead search—can significantly outperform PRM-free approaches like Majority@N. In contrast, our study finds that in the **Long CoT model** setting, beam search does not consistently outperform Majority@N.
>
> We acknowledge this difference, but we believe the conclusions are not contradictory. The prior work is based on non-Long CoT models with relatively short reasoning traces, whereas our study focuses on PRM behavior under the Long CoT model setting, where the extended reasoning depth and structure present different challenges.
>
> To further test the generality of our proposed ZGES method and to better reconcile these observations, we follow the reviewer's suggestion and evaluate the standard lookahead search method described in *Scaling LLM Test-Time Compute Optimally* under the Long CoT model setting. We then apply ZGES on top of it for comparison. **The results are as follows.**
>
> | Model | Method               | AMC23         | AIME24        | AIME25        |
> |:-----:|:---------------------|:--------------|:--------------|:--------------|
> |**1.5B**       | 🔴 **ZGES (λ = -0.4)** | **0.910** ↑2.5% | **0.600** ↑7.4% | **0.366** ↑2.0% |
> |**1.5B**       | 🔴 **ZGES (λ = -0.8)** | **0.915** ↑3.0% | **0.580** ↑5.4% | **0.373** ↑2.7% |
> |**1.5B**| Look Ahead Search          | 0.885          | 0.526          | 0.346          |
> |       |                      |                |                |                |
> |**7B**        | 🔴 **ZGES (λ = -0.4)** | **0.945** ↑0.5% | **0.780** ↑3.3% | **0.574** ↑6.1% |
> | **7B**       | 🔴 **ZGES (λ = -0.8)** | **0.945** ↑0.5% | **0.767** ↑2.0% | **0.573** ↑6.0% |
> |**7B** | Look Ahead Search          | 0.940          | 0.747          | 0.513          |
>
> > **Note:**
> > - *Look Ahead Search* refers to the standard PRM-Guided Look Ahead Search without early stopping.
> > - *ZGES(x)* denotes the ZGES method (used for Look Ahead Search) with λ set to x.
>
> The experimental results are averaged over different beam size settings, with the expansion size fixed at 2. The results are highly encouraging: the ZGES method significantly outperforms the standard look-ahead search, suggesting that:
>
> 1. **ZGES demonstrates strong generalizability** and can adapt to more general search algorithms.
> 2. **Under the Long CoT model setting**, early stopping proves beneficial for both Beam Search and Look-Ahead Search, indicating that our observation may reflect a more general pattern.
>
>
> ---
>
> >**Weakness 2. Lack of PRM training and accuracy analysis**
>
> Thank you for pointing this out. We agree that understanding the fidelity of our trained PRM is critical. The training details of our PRM are provided in `Appendix B.1.` Our training procedure closely follows the method proposed in *Math-Shepherd: Verify and Reinforce LLMs Step-by-Step Without Human Annotations*.  To ensure the reliability of our PRM, we followed the reviewer’s suggestion and evaluated our trained PRM on PRMBench.  However, due to time constraints and the rebuttal deadline, the evaluation has not yet been completed. Once it is finished, we will promptly update the results and share them with you.
>
>
> ---
>
> >**Weakness 3. Limited domains (only math reasoning)**
>
> We thank the reviewer for pointing this out. You are absolutely right that evaluating ZGES on tasks beyond mathematical reasoning is important to demonstrate its generalization capability. Following your suggestion, we conducted experiments on the GPQA dataset, comparing the standard PRM-Guided Beam Search and our proposed ZGES method. The results are shown below:
>
> | Model | Method               | GPQA         |
> |:-----:|:---------------------|:--------------|
> |**7B**        | 🔴 **ZGES (λ = -0.4)** | **0.547** ↑2.7% |
> | **7B**       | 🔴 **ZGES (λ = -0.8)** | **0.541** ↑2.1% |
> |**7B** | Beam Search         | 0.520          |
>
> > **Note:**
> > - *Beam Search* refers to the standard PRM-Guided Beam Search without early stopping.
> > - *ZGES(x)* denotes the ZGES method (used for Beam Search) with λ set to x.
> ---
>
> >**Weakness 4. Missing details on step segmentation**
>
> We thank the reviewer for the insightful comment. In general, step segmentation can be based on symbolic cues (e.g., explicit "Step" annotations or line breaks), model confidence (see [1]), or token-based heuristics. Given that DeepSeek-R1-Distill-Qwen models do not yield outputs with clearly demarcated step boundaries, we adopt a hybrid segmentation strategy that integrates symbolic markers and token-length priors. Specifically, we treat certain tokens—such as line breaks or step-indicative phrases—as potential split points, and further apply this rule approximately every 2000 tokens to prevent overly fragmented segmentation in long outputs. This approach yields around 10–20 steps per output in practice and works well empirically.
>
> [1]: *AdaptiveStep: Automatically Dividing Reasoning Steps through Model Confidence, ICML 2025.*
>
>
> ---
>
> >**5. Minor typo**
>
> Thank you for pointing this out. We appreciate your careful reading and have corrected the typo ("reasonoing" → "reasoning") in line 87. The correction will be reflected in the revised version of the paper
>
> ---
>
> **Summary**
>
> We believe the reviewer’s comments have helped us clarify our methodology, better contextualize our contributions with respect to prior work, and strengthen the empirical evaluation. We hope the revised analysis addresses the concerns raised.

---

> > ### Comment · Reviewer_egya · 2025-08-02
> >
> > Thank you for the author’s reply. Your response has addressed part of my concerns; however, the two core questions remain insufficiently answered.
> >
> > First, the reply lacks theoretical justification or a more detailed analysis explaining why there is such a large performance gap when applying the proposed method to long-CoT and short-CoT settings, or to AlphaGo-style algorithms. Moreover, the author did not address the part of my first question regarding conducting experiments on more existing PRMs.
> >
> > Second, the author merely acknowledged the issue but did not provide any preliminary attempts. I believe this would not take much time, as it only requires running the evaluation on the test set and the computational cost should be modest.
> >
> > These two questions are central to both the method and its effectiveness, so I believe it is necessary to address them thoroughly.

---

### Decision · Program_Chairs · 2025-09-17

**Decision:**

Accept (poster)

**Comment:**

The paper identifies and analyzes a deficiency of process reward model (PRM)-based beam search, showing that it arises from the degradation of PRM quality as the search step increases. To address this issue, the authors propose a Z-score guided early stopping algorithm and demonstrate its performance benefits. The contribution is timely and interesting. Some concerns, as noted by the reviewers, remain. For example, comparisons with additional existing PRM baselines would strengthen the work. Overall, the strengths outweigh the weaknesses, and I recommend acceptance. I also encourage the authors to incorporate the reviewers’ comments into the camera-ready version.